# New Insights into TETs in Psychiatric Disorders

**DOI:** 10.3390/ijms23094909

**Published:** 2022-04-28

**Authors:** Wenxia Zhan, Yi Li, Jie Yuan, Na Zhi, Yiyuan Huang, Yingqi Liu, Ming Zhang, Shengxi Wu, Xianghui Zhao

**Affiliations:** 1Department of Neurobiology, School of Basic Medicine, Fourth Military Medical University, Xi’an 710032, China; wxzhan1526@163.com (W.Z.); lyliyi0@163.com (Y.L.); yuanjie01042021@163.com (J.Y.); zhina040922@163.com (N.Z.); eguorpmal@163.com (Y.H.); yingqiliu0204@163.com (Y.L.); uknow0206m@sina.com (M.Z.); 2College of Life Sciences, Northwest University, Xi’an 710127, China; 3School of Basic Medicine, Yan’an University, Yan’an 716000, China

**Keywords:** psychiatric disorder, TET enzyme, 5-hydroxymethylcytosine

## Abstract

Psychiatric disorders are complex and heterogeneous disorders arising from the interaction of multiple factors based on neurobiology, genetics, culture, and life experience. Increasing evidence indicates that sustained abnormalities are maintained by epigenetic modifications in specific brain regions. Over the past decade, the critical, non-redundant roles of the ten-eleven translocation (TET) family of dioxygenase enzymes have been identified in the brain during developmental and postnatal stages. Specifically, TET-mediated active demethylation, involving the iterative oxidation of 5-methylcytosine to 5-hydroxymethylcytosine and subsequent oxidative derivatives, is dynamically regulated in response to environmental stimuli such as neuronal activity, learning and memory processes, and stressor exposure. Here, we review the progress of studies designed to provide a better understanding of how profiles of TET proteins and 5hmC are powerful mechanisms by which to explain neuronal plasticity and long-term behaviors, and impact transcriptional programs operative in the brain that contribute to psychiatric disorders.

## 1. Introduction

Psychiatric disorders are multifactorial disorders with brain dysfunctions manifested at different degrees of perceptive and cognitive deficiency, such as autism spectrum disorders (ASDs), schizophrenia (SCZ), major depression (MD), and anxiety disorders. Previous research indicates that epigenetic mechanisms play an important role in the etiology of psychiatric conditions [1,2]. As the most studied form of epigenetic modifications, DNA methylation is maintained throughout life and is involved in transposon silencing, X-chromosome inactivation and genomic imprinting [3]. It is generally believed that DNA methylation represses gene expression [4] and participates in the generational transmission of environmentally acquired traits in a variety of psychiatric disorders [5].

DNA methyltransferase (DNMT) catalyzes the selective addition of a methyl group to the fifth carbon on the cytosines of CpG dinucleotide, namely 5-methylcytosine (5mC) [6]. There are four types of DNMT in mammals, but only the following three show catalytic activity in humans: DNMT1, DNMT3a and DNMT3b. DNMT1 is responsible for transmitting DNA methylation information to offspring and maintaining the existing 5mC during DNA replication and repair, while DNMT3a and DNMT3b establish dynamic methylation patterns during development [7]. In 2009, DNA hydroxymethylase, ten-eleven translocation proteins (TETs), was identified to convert 5mC into 5-hydroxymethylcytosine (5hmC) [8]. Furthermore, 5hmC is the first oxidative product in the cytosine demethylation pathway and is plenteous in the central nervous system (CNS) [9,10]. The name of TET derives from the fact that the human *TET1* gene was originally identified in rare cases of acute myeloid leukemia, participating in the ten eleven translocation [t(10;11) (q22;q23)] related to the MLL gene [11]. Both TET enzymes and 5hmC have shown crucial functions in the regulation of various neural activities and pathological processes of the brain [12], suggesting new discoveries in the interaction between genetic and environmental clues. Although the role of DNA methylation in neuronal disorders has been studied for decades, the function of DNA demethylation remains unclear. In this review, we summarized the experimental and clinical findings about TET enzymes and 5hmC in psychopathic disorders and discussed how such epigenetic alterations may interfere with neurodevelopment and contribute to the variation of pathological traits and provide a better understanding of novel treatments.

For the function of TETs and 5hmC in psychiatric diseases, a systematic literature survey was conducted using “psychiatric disorders”, “TET enzyme”, “5-hydroxymethylcytosine”, “autism spectrum disorders”, “schizophrenia”, “major depression” and “anxiety disorders” as key words to search the peer-reviewed publications between the year of 2010 and 2022, in the following international reference resources: Public Medline (PubMed; National Institutes of Health, https://pubmed.ncbi.nlm.nih.gov/, (accessed on 20 February 2022)), Europe PubMed Central (EuroPMC; European Bioinformatics Institute, https://europepmc.org/, (accessed on 20 February 2022)). Any publications that were not written in English were excluded from our review.

## 2. Dual Functions of TET in Transcriptional Regulation

### 2.1. TET Structure and Catalytic Activity

The human TETs are long proteins, with 2039 amino acids (aa) in TET1, 1921 aa in TET2, and 1803 aa in TET3 [10]. All three members consist of a cysteine-rich region at the N-terminus and a double-stranded β-helix domain (DSBH) at the C-terminus [13]. The catalytic domains of all three TETs are situated in C-termini [8]. The DSBH domain is responsible for iron binding and is associated with the oxygenase activity of TET proteins [13,14]. A sequence analysis revealed high similarity between TET1 and the C-terminal domain of *Saccharomyces cerevisiae* RNA polymerase II, which regulates RNA polymerase II activity through protein methylation, suggesting a role of TET1 in regulating post-translational modifications [15]. Different from TET2, both TET1 and TET3 contain a DNA-binding zinc finger cysteine-X-X-cysteine (CXXC) domain in their N-terminus, which is thought to recognize unmethylated CpGs and is essential for their functions [16]. However, due to a chromosomal inversion, TET2 does not contain a CXXC domain, which is compensated for its neighboring gene, the “inhibition of the Dvl and AXin complex” (*IDAX*), and is responsible for recruiting TET2 to DNA [17]. The structure of TET proteins is demonstrated in Figure 1.

TET proteins are the enzyme family of 2-oxoglutarate-dependent dioxygenases (2-OGDDs) that the oxygenase activity depends on, with Fe^2+^, 2-OG, molecular oxygen, and vitamin C as reducing agents [18,19]. Their common reaction process is as follows: in the presence of a reducing cofactor (generally Fe^2+^), the substrate is hydroxylated by O_2_ and the 2-OG co-substrate is decarboxylated to succinate and CO_2_ [20]. Natural or synthetic 2-OG analogues have been identified to inhibit TET dioxygenase activity [21,22]. Tumor-associated succinate, fumarate and R-2-hydroxyglutarate (R-2HG) have been proven to repress catalytic activity for TET1 and TET2, but not TET3. Among them, fumarate has the strongest inhibitory effect, followed by succinate [23]. Gene mutations for succinate dehydrogenase, fumarate hydratase and isocitrate dehydrogenase have been identified in several cancers, leading to the amassment of the 2-OG analogs succinate, fumarate and R-2HG, respectively [24]. This corresponds to the reduced activity of TETs in relevant tumors [5]. Meanwhile, except for Fe^2+^, divalent metals can also inhibit the oxygenase activity of 2-OGDDs by competing with Fe^2+^ [25]. Vitamin C contributes to the oxygenase function of TET, but it is not essential [26].

### 2.2. Non Oxygenase Activity of TETs

Although the main function of the TET protein was considered to be oxygenase activity dependent gene expression regulation, emerging studies have determined the importance of the non-catalytic function of TETs [18,27] (Figure 2). For example, TET1 can promote the expression of *srGAP3*, independent of its enzymatic activity, and negatively regulates the neuronal differentiation of Neuro2a cells [28]. Recent research using TET2 catalytically inactive mice showed the significance of the non-catalytic role of TET2 in the hematopoietic system [29]. Besides, the essential role of TET3 in maintaining the adult neural stem cell (NSC) pool is believed to be non-catalytic and acts through its direct binding to the target gene, Small ribonucleoprotein-related polypeptide N (*Snrpn*) [30]. Furthermore, studies have established interaction between TETs-mediated epigenome and O-linked N-acetylglucosamine (O-GlcNAc) transferase (OGT)-mediated metabolism [31,32,33,34]. The catalytic domain of TET2 can directly interact with OGT, which is required for histone O-GlcNAcylation and is unrelated to TET2 oxygenase activity [31]. Meanwhile, the interaction between TET1 and OGT can promote TET1 activity during development and disease [32,33,34]. These observations indicate the complex dual activities of TETs in regulating gene expression and can be implicated in designing strategies for targeting TETs in disease treatment.

## 3. TETs-5hmC Function in Nervous System

Since 2009, with the development of novel chemical detecting methods, many studies have shown that 5hmC is highly distributed in the brain and spinal cord. Kriucionis and Heintz found that 5hmC accounted for about 0.6% and 0.2% of total nucleotides in Purkinje neurons and the granulosa cells of mice brain, using high pressure liquid chromatography (HPLC), thin layer chromatography (TLC) and mass spectrometry (MS) [9]. Subsequently, it was found that in the cortex and hippocampus, the amount of 5hmC is much greater, reaching 0.7% [35]. Similarly, with a HPLC-MS analysis and immunohistochemistry assay, Globisch and co-workers discovered that the concentration of 5hmC in the CNS was the highest among all tissue types [36]. Alexey Ruzov also confirmed this observation and suggested that a high 5hmC level is characteristic of neuronal progenitors and mature neurons [37]. These observations indicate the critical role of TET-5hmC in regulating brain formation and functions.

### 3.1. TETs in Neurogenesis

The differentiation of embryonic stem cells (ESCs) requires large-scale histone and DNA modification. A study using TET1-3 triple-knockout (TKO) human ESCs shows defective neuroectoderm differentiation and prominent PAX6-promoter hypermethylation, which is a key regulator in neuroectoderm development [38]. Consistently, another study showed a decreased 5hmC level in PAX6 promoter after TET1 knockdown in hESCs, which induced a lower expression of PAX6 and inhibited the differentiation of hESCs into neuroectoderm [39]. In addition, it has been observed that *Tet3* is upregulated and *Tet1* is downregulated during the differentiation of mouse ESCs to NPCs [40]. The knockdown of *Tet3* in NPCs leads to a concomitant hypomethylation of pluripotency-associated genes, suggesting the role of TET3-5hmC in silencing pluripotency genes and consequently maintaining stem identity in NPCs.

Studies also suggested that TETs were highly relevant to adult hippocampal neurogenesis. Down syndrome critical region 1 (DSCR1) protein knockout mice have been found to exhibit impaired neurogenesis and increased TET1 expression in adult hippocampus [41]. Correcting the TET1 level in *Dscr1* knockout mice reduces the demethylation of miR-124 promoter and achieve the optimum level of miR-124, thereby restoring defects in adult hippocampal neurogenesis. Likewise, *Tet1* overexpression affects adult hippocampal neurogenesis and *Tet1*-transgenic mice showed elevated anxiety and enhanced fear memories [42]. In addition, *Tet1* overexpression activates calcium signals in excitatory neurons and increases the expression of immediate early genes, especially in the prefrontal cortex and hippocampus. Evidence from fetal growth restriction (FGR) mice that exhibit impaired learning and memory ability has suggested that a reduced level of *Tet1* in hippocampal NSCs may induce hypermethylation and the downregulation of Notch genes, and block the proliferation of NSCs [43]. These changes ultimately impair developmental neurogenesis in the hippocampus. Aside from TET1, the knockdown of *Tet2* in the hippocampus leads to decreased neurogenesis and an impaired cognitive ability [44]; overexpression of *Tet2* induces the opposite effect. These observations indicate that TET proteins are critical molecular mediators for neurogenic rejuvenation and it may provide a feasible approach to alleviate the precipitous aged-related decline in neurogenesis.

### 3.2. Role of TETs and 5hmC in Learning and Memory

Since TETs are involved in adult neurogenesis, evidence has suggested that TET proteins can regulate learning and memory formation by modulating the dynamic level of neurodevelopment related genes. Garrett A. Kaas et al. found that TET1-5hmC can promote the expression of genes associated with learning and memory, including *Npas4*, *c-Fos*, and *Arc*, and that the overexpression of *Tet1* in the hippocampus impairs memory formation [45]. Another group reported that physical exercise can rescue the age-dependent decrease of TET1 and TET2 levels in the hippocampus, increase the 5hmC level of the miR-137 internal promoter, and significantly improve spatial memory in aged mice [46]. Additionally, TETs can also be guided to target genes by binding patterners and can regulate brain epigenetic programming and memory formation. For example, TET1 interacts with EGR1 and is recruited to EGR1 target genes, thereby activating the expression of downstream genes through DNA demethylation and regulating memory formation [47].

A recent study showed that *Tet1* gene is expressed in two isoforms during development [48]. The N-terminally truncated TET1 protein, encoded by a shorter *Tet1* transcript (*Tet1-*S), is highly enriched in neurons, and a full-length *Tet1* transcript (*Tet1-*FL) is much more abundant in glia. Interestingly, suppression of each individual TET1 isoform results in distinctive changes in neuronal gene expression and memory formation. In particular, different from *Tet1*-FL, the acute repression of *Tet1*-S results in a higher expression of synapse-associated genes [48].

Except for their physiological role, TET enzymes have also been shown to regulate neuronal death and repair. For example, cerebellar granule cells with *Tet1* deletion are sensitive to oxidative-stress-induced neuronal apoptosis, which can be remarkably rescued by the overexpression of the catalytically active domain of TET1 (TET1-CD) [49]. This effect is DNA demethylation-dependent and is related to the expression of the *Klotho* gene in neurons. Besides, there is evidence showing that TET-mediated DNA demethylation can permit axonal regeneration in adult mice. For example, *Tet1* knockdown in retinal ganglion neurons attenuated *Pten* ablation-induced axon regeneration in adult mice [50,51]. Additionally, upon sciatic nerve lesion, *Tet3* expression is upregulated in DRG neurons and active DNA demethylation induces the expression of multiple regeneration-associated genes [50]. This observation indicates that TET3 is necessary for neural repair and behavioral recovery after injury.

### 3.3. TET-5hmC in Glia Biology

In addition to neuronal functions, many works have identified the essential function of TETs and 5hmC in glia biology. Carrillo-Jimenez et al. found that TET2 expression is increased in microglia upon exposure to various inflammogens [52]. Then, TET2 mediated early gene transcriptional changes can lead to metabolic alterations and inflammatory responses later on. This effect is independent of TET2 catalytic activity. In oligodendrocytes (OLs) that enwrap axons and assist in the saltatory conduction of action potentials in the CNS, TET1-5hmC has been identified to be necessary for both myelin formation and repair [53,54]. This effect is related to the TET1-5hmC regulated epigenetic program and intracellular calcium activity in OLs. Similarly, another study revealed that in the young adult mice, TET1 is the main enzyme catalyzing 5hmC formation in OLs [55]. The level of TET1 declines with age, which impacts the expression of genes regulating neuro-glial communication, e.g., *Slc12a2*, thus causing inefficient remyelination in the old mice. These observations suggest the fundamental role of TET1 in OL-associated functions, and provides therapeutic strategies for demyelinating disease.

In drosophila, the single homologous *Tet* gene, *dTet*, is expressed in a specific type of glia, midline glia (MG). MG is involved in the regulation of axon connectivity and is required for proper axonal development in the ventral nerve cord [56]. Knockdown of *dTet* represses the expression of axon guidance genes, such as *Slit*, and impairs axon patterning in the ventral nerve cord. These morphological defects finally cause locomotor dysfunction and increase lethality. Further investigations are required to understand the link between a loss of TETs activity in glia and neuronal functions, such as synaptogenesis and neurite formation.

## 4. TET-5hmC in Neuropsychiatric Disorders

The essential role of TET-5hmC in brain development makes them promising plausible molecular sources for neuropsychiatric disorders, especially with developmental origins, such as ASD, SVZ and BP [57]. For example, studies show that the mRNA and protein level of TET1 in the inferior parietal lobule (IPL) of psychotic patients is much higher than that of non-psychotic patients, along with an increased level of 5hmC [58]. Further elucidation of their involvement in the neurobiological underpinnings of psychiatric disorders may deepen our understanding of these disease and may provide optimal therapeutic options through epigenetic interventions in the future.

### 4.1. TET1-5hmC in Autism Spectrum Disorders

Autism spectrum disorders belongs to developmental disorders that are characterized by social withdrawal, repetitive behaviors and limited interests [59]. It includes several disorders that share common features but are different to each other, such as autistic disorder, Rett syndrome, Asperger’s syndrome, pervasive developmental disorder not otherwise specified (PDD-NOS), and childhood disintegrative disorder [60].

Despite the fact that etiologies for ASD remain elusive, various studies suggest that altered methyl-CpG binding protein 2 (MeCP2) levels due to gene deletions or mutations are highly relevant to this spectrum of neurodevelopmental disorders, especially in Rett’s disorder and MeCP2 Disorders [61,62,63]. As a reader for DNA methylation, MeCP2 binds to modified CpG dinucleotides with high affinity and its function is highly relevant to cytosine modifications. The study of Mellen et al. identified MeCP2 as one of the main 5hmC binding proteins in the CNS, which promotes gene transcription when binding to 5hmC, and vice versa when binding to 5mC-containing DNA [64]. In the cerebellar cortex of ASD patients, the investigators find a significantly increased level of MeCP2 binding to the promoters of *GAD1* and *RELN* genes [65]. Additionally, this may relate to the marked upregulation of TET1 expression in ASD patients, the enrichment of 5hmC levels at *GAD1* and *RELN* promoters, and an increased binding of TET1 to these promoter regions.

There is further evidence supporting the function of 5hmC mediated epigenetic regulation in the pathogenesis of ASD. Ligas et al. generated a genome-wide disruption map of 5hmC in the striatum from a mouse model of autism (*Cntnap2* KO mice), which revealed that compared to control mice, many differentially hydroxymethylated regions (DhMRs) were annotated in genes that ontologically function in axonogenesis and neural morphogenesis, e.g., *Nlgn* and *Reln* [66]. Besides, a recent study using DNA samples from the postmortem brain of ASD patients identified age-dependent DhMRs that are highly associated with intercellular communication and brain disorders [67]. Together, these results represent a critical improvement towards an understanding of the biological complexity of autism pathogenesis and suggest the TET-5hmC modulated epigenetic program as novel target to shape the molecular consequence of autism phenotypes.

### 4.2. TET1-5hmC in Schizophrenia

Schizophrenia (SCZ) is a chronic, life-long, and debilitating disorder characterized by the presence of psychosis, social withdrawal, unusual and uncharacteristic behavior, and cognitive impairment, which could have a genetic and neurobiological background. The worldwide prevalence of this disease is about 1%. A considerable number of studies have indicated that epigenetic changes are highly associated with the onset of SCZ. A pathological study on brains of SCZ patients identified elevated expression levels of TET1 mRNA and protein in PFC [68]. Meanwhile, the 5hmC level in SCZ-related genes, such as glutamic acid decarboxylase 67 and brain-derived neutrophic factor *(BDNF*), increased in their promoter regions [69,70]. In addition, a higher TET1 expression was found in hippocampus of SCZ mice model [71], and the REST-regulated internal promoter of miR-137, a candidate gene for SCZ, is highly hydroxymethylated [46,72]. Although there is still considerable uncertainty when using peripheral tissues as biomarkers for neuropsychiatric disorders, especially blood and saliva, a study using blood from first-episode psychosis patients and healthy controls by ELISA assay revealed the correlation between increased expression of TET1, NRG1, BDNF, DNMT1 and the occurrence and development of SCZ [73]. Our recent study revealed that TET1-5hmC dysfunction in OLs weakens the sensorimotor gating ability and impairs cognitive ability in mice [53], which are considered the core symptoms of SCZ. This observation suggests an important role of glia-mediated epigenetic regulation in SCZ etiology.

Compared to the knowledge of DNA methylation in SCZ, the understanding of TET-5hmC-mediated epigenetic mechanism underlying schizophrenia remains at the onset stage. Further elucidation of the interactions between genetic risk alleles and environmental insults on target gene 5hmC modification will provide new molecular insights for understanding the etiology.

### 4.3. TET-5hmC and Depression

Depression is a mood disorder characterized by persistent low mood, loss of interest, and decreased energy [74]. Major depressive disorder (MDD), the most severe form of depression, is a psychiatric disorder that affects one in five people’s quality of life and is one of the leading causes of the global disease burden [75]. The epigenetic mechanism, which provides a basis for the integration of genetic and environmental factors, is suggested to explain gene–environment interactions in the pathogenesis of depression [76].

As to the dynamics of DNA methylation in MDD patients, there are contradictory results in the research using DNA genomes extracted from the blood, which may come from different detection methods [77,78,79]. Interestingly, the alteration of both 5mC and 5hmC seems to be more obvious in the elderly group, suggesting that changes in DNA methylation are age-related in patients with major depression [79]. Recently, genome-wide 5hmC profiling has been detected by AbaSI sequencing in the PFC of patients with depression [80]. Although the distribution of 5hmC did not show significant difference between depressed and psychiatrically healthy groups, the authors identified DhMRs in CpGs located in genes encoding *myosin XVI* and *insulin-degrading enzymes*. They verified that the changes in the expression of these genes were associated with the enrichment of 5hmC in depressed suicides [80].

In an ethologically validated mouse model of depression (chronic social defeat stress), the TET1 level was reduced in stress-susceptible mice, especially in nucleus accumbent (NAC) [81]. Moreover, *Tet1* knockout in NAC neurons leads to antidepressant- and anxiolytic-behavior phenotypes in adult mice. In addition, Benoit Labonté et al. found that the DNA methylation level of the growth arrest and DNA damage-inducible 45 (*Gadd45*) gene is associated with stress susceptibility in mice, thereby causing depression-like behaviors [82]. Furthermore, Gadd45 interacts with thymine DNA glycosylase (TDG) and promotes DNA demethylation, as well as gene transcription through a TDG-TET dependent way [83]. Therefore, whether the contribution of Gadd45 in depression disorder involves TET-5hmC axis needs further exploration.

### 4.4. TET3 and 5hmC in Anxiety Disorders

Anxiety disorders are neuroses characterized by anxiety, including generalized anxiety, panic disorder and phobia [84]. Patients suffering from anxiety are usually accompanied by hypothalamic-pituitary-adrenocortical (HPA) axis hyperresponsiveness [85], and the role of TET3 in regulating HPA axis and neuronal activity has been illustrated [86]. Antunes and co-workers identified that TET3 is required for maintaining the proper anxiety level and normal spatial cognitive function [86]. They notice that *T**et3* deletion in mature neurons induces a dysregulation of genes related to the HPA axes, which are generally regarded as susceptibility-related genes for anxiety, and increases IEGs’ expression in the hippocampus. Besides, the corticotropin-releasing hormone (CRH) is vital in determining HPA axis-related stress adaptation [87], which can be demethylated by TET3 to activate transcription under different heat stress conditions [88]. This effect is accomplished by TET3 integration into a protein complex containing the RE1 silencing transcription factor that targets CRH intron, and is suggested to determine the resistant or susceptible response to stress in later life.

Besides, a genome-wide 5hmC profiling for hypothalamus from adult female mice revealed that anxiety-like behaviors were associated with several potential stress-related genes [89]. These genes revealed a disrupted 5hmC level and are involved in neurodevelopment and differentiation. These findings provide a substantial basis for further revealing the molecular mechanisms of anxiety disorders.

## 5. Conclusions

In this review, we attempted to present novel findings of TET proteins in brain function and dysfunction. During neurodevelopment, TETs coordinate the formation of primitive CNS structures, neurogenesis, and neuroplasticity, and studies using loss or gain-of-function strategies have suggested non-redundant functions for TET enzymes in the brain. However, a full understanding of how TETs regulate neuronal physiology, plasticity and cognitive functions is still a matter of debate and requires further investigation. We consider this review to be a snapshot of where things stand at this point in time, with emerging new genetic and epigenetic mechanisms and targets surely just over the horizon. Considering the effect of TETs in various neuropsychiatric disorders (As summarized in Table 1), further investigation is needed to fully clarify the expression of each TET enzymes during typical neurodevelopment and the normal levels of variance in neurotypical subjects. In addition, the catalytic and non-catalytic functions of TET proteins in regulating gene expression have not been fully defined in multiple biological processes, which are required to reveal the heterogeneous etiology and pathophysiology of neuropsychiatric disorders, and may also facilitate attempts to identify better therapeutic strategies for these diseases.

## Figures and Tables

**Figure 1 ijms-23-04909-f001:**
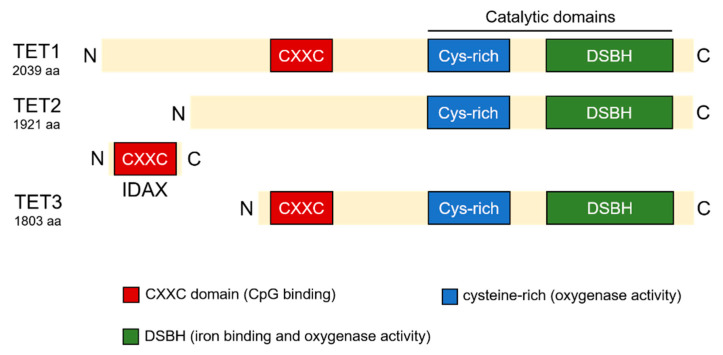
The structure of TET proteins.

**Figure 2 ijms-23-04909-f002:**
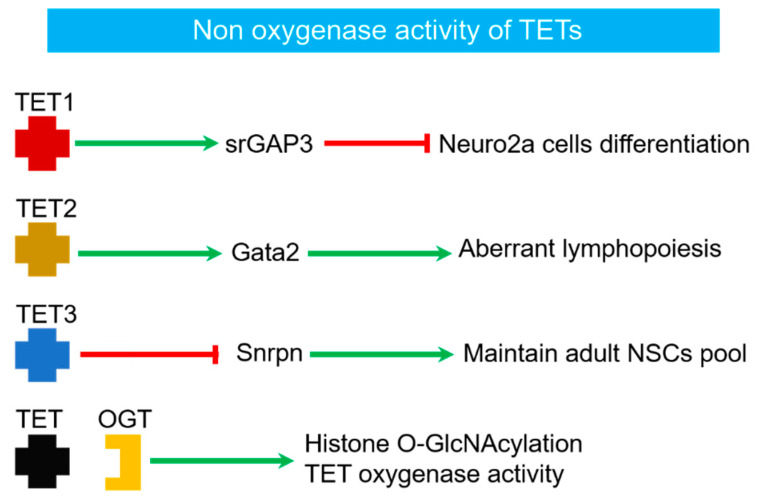
Examples of Non oxygenase activity of TET proteins.

**Table 1 ijms-23-04909-t001:** TETs and 5hmC in different psychiatric disorders.

Type of Psychiatric Disorders	Alterations in TETs/5hmC	Downstream Mechanisms/ Behavior Effects	Brain Region/Cell Type	Sample	References
ASD	Increased TET1 expression, increased level of 5-hmC at the promoters of GAD1 and RELN, increased TET1 binding to target gene promoter regions	A significant increase in MeCP2-binding to the hyperhydroxymethylated promoter regions of GAD1 and RELN	Cerebella	ASD postmortem	[65]
	Genome-wide 5hmC level decrease, mainly in gene regions and repetitive elements	DhMRs annotation revealed a significant overlap with known ASD genes (e.g. Nrxn1 and Reln)	Striatal	*Cntnap2*^−/−^ mice model for ASD	[66]
	Specific set of DhMRs in young age group (<18)	Significant overlap between DhMRs-related genes and ASD risk genes	Cerebella	ASD postmortem	[67]
SCZ	Increased TET1 level	Higher DNA demethylation at GAD67 and BDNF promoter regions	Prefrontal cortex	SCZ postmortem	[68,69,70]
	Increased TET1, together with increased DNMT1 and decreased NRG1, ErbB4, BDNF	Combination of ErbB4, BDNF and TET1 as biomarkers for SCZ diagnosis	Blood samples	SCZ patients	[73]
	TET1 function deficiency/Genome-wide 5hmC decrease, especially in gene body region	Hyper-hydroxymethylation level in myelination genes, cell cycle genes and calcium transporter genes/SCZ like behavior	Oligodendrocyte lineage cells	*Tet1* conditional knock out mice	[53]
Depression	Genome-wide decreased 5-hmC level	5-mC levels positively correlated with severity of depressive symptoms	Blood samples	BD or MDD patients	[77]
	Significant decrease of 5-hmC level in older age group	MDD may curtail the rise in methylation levels during normal aging	Leukocyte in blood sample	MDD patients	[79]
	Increased 5hmC level in genes encoding myosin XVI and insulin-degrading enzymes	Target genes are abnormally expressed in depressed suicides	Prefrontal cortex	Depression postmortem	[80]
	*Tet1* ablation	Produced antidepressant-like effects	NAc neurons	Selective *Tet1* knockout	[81]
	Decreased TET1	TET1 negatively regulates reward behavior in the NAc through extensive dynamic changes in 5hmC at response genes	NAc neurons	CSDS mouse model for depression	
Anxiety disorders	*Tet3* ablation	Dysregulated genes involved in glucocorticoid signaling pathway (HPA axis) and upregulation of immediate early genes in hippocampus/Increased anxiety-like behavior	Adult brain neurons	*Tet3* conditional knock out mice	[87]
	Increased expression of TET3	TET3-REST (silencing transcription factor) binary complexes to CRH intron/Improve the stress response late in life	Hypothalamic paraventricular nucleus	Experienced acute heat stress mice model	[88]
	Disrupted 5hmC	Disrupted gene expression in stress-related targets (eg. Nr3c2, Nrxn1, Nfia, and Clip1)/Anxiety-like behaviors in adult female mice	Hypothalamus	Experienced early-life stress mice model	[89]

Abbreviations: MeCP2: methyl-CpG binding protein 2, GAD1: glutamate decarboxylase 1, RELN: reelin, DhMRs: differentially hydroxymethylated regions, Nrxn1: Neurexin 1, GAD67: glutamic acid decarboxylase67, BDNF: brain-derived neutrophic factor, NRG1: Neuregulin1, DNMT1: DNA methyltransferases 1, BD: bipolar disorder, MDD: Major depressive disorder, NAc: nucleus accumbens, CSDS: chronic social defeat stress, CRH: corticotropin-releasing hormone, REST: silencing transcription factor.

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
