# Peer review of "New Insights into TETs in Psychiatric Disorders"

_ijms, 2022, doi:10.3390/ijms23094909_

Round 1

Reviewer 1 Report

The present review focuses on studies about TET proteins and 5hmC in order to deepen our knowledge on mechanisms of neuronal plasticity and impact on transcriptional programs in the brain that contribute to psychiatric disorders.

I've found this study interesting and up-to-date. 

I have two suggestions that might improve the readability of the mauscript:

  • add a methods section (how literature search was conducted, inclusion/exclusion criteria etc.)
  • add a table showing main results in the different psychiatric diseases (schizophrenia, autism spectrum, depression etc.) 

Author Response

We agree with the reviewer’s suggestions and have included a section introducing how literature research was conducted (page 2) and added a new table showing main results in different psychiatric diseases in the revision.

Reviewer 2 Report

This review paper discusses the role of the ten-eleven translocation proteins (TETs), in the development of the brain, which may be linked to the etiology of psychiatric disorders. The discussed topic is very important, however the quality of the paper would improve, if the Authors introduce the following corrections:

When you write a review paper, you need to describe the methodology of your search strategy. You should list the databases you used and describe the search terms. For example you should use the elements of the PICO model: Patient/ Problem, Intervention, Comparison and Outcome.

I would be also valuable to create a table showing the main search results.

It is widely recommended to use PRISMA guidelines for reviews.

There also some minor issues that need a correction:

- line 30: There are different anxiety disorders, so I suggest not to use singular form "anxiety disorder"

- line 31: In English language It is recommended to use singular form "research" and not "researches"

Author Response

Q: When you write a review paper, you need to describe the methodology of your search strategy. You should list the databases you used and describe the search terms. For example you should use the elements of the PICO model: Patient/ Problem, Intervention, Comparison and Outcome.

A: We agree with the reviewer’s suggestion for adding the methodology of our search strategy and have list the databases and inclusion/exclusion criteria in the methods section (page 2).

Q: I would be also valuable to create a table showing the main search results.

A: We thank the reviewer’s suggestion. To improve the readability of our manuscript, we have created a new table showing the main results in different psychiatric disease.

Q: It is widely recommended to use PRISMA guidelines for reviews.

A: We think the PRISMA guidelines are designed for systematic reviews and our submission belongs to literature/narrative review, to which the PRISMA guidelines cannot be applied. Also, we have checked several literature review publications in International Journal of Molecular Sciences (https://doi.org/10.3390/ijms23031923; https://doi.org/10.3390/ijms23084264; https://doi.org/10.3390/ijms23084306; https://doi.org/10.3390/ijms23084288), and all these reviewers did not use PRISMA guidelines.

Q: There also some minor issues that need a correction:

- line 30: There are different anxiety disorders, so I suggest not to use singular form "anxiety disorder"

- line 31: In English language It is recommended to use singular form "research" and not "researches"

A: We have corrected the use of these two words: “anxiety disorders” (line 30 and line 296) and “research” (line 31, line 219, line 224, line 275).

Round 2

Reviewer 2 Report

I would like to thank the Authors for all changes and improvements they have done. I have no further comments.